# Mouse-Derived Isograft (MDI) In Vivo Tumor Models I. Spontaneous sMDI Models: Characterization and Cancer Therapeutic Approaches

**DOI:** 10.3390/cancers11020244

**Published:** 2019-02-19

**Authors:** Peter Jantscheff, Janette Beshay, Thomas Lemarchand, Cynthia Obodozie, Christoph Schächtele, Holger Weber

**Affiliations:** 1In Vivo Pharmacology, ProQinase GmbH, Breisacher Str. 117, 79106 Freiburg, Germany; Janette_Beshay@web.de (J.B.); c.obodozie@proqinase.com (C.O.); c.schaechtele@proqinase.com (C.S.); 2TPL Pathology Labs, Sasbacher Str. 10, 79111 Freiburg, Germany; lemarchand@tpl-path-labs.com

**Keywords:** mouse tumor models, experimental cancer, mouse-derived isografts (MDIs), spontaneous tumors, carcinogen-induced tumors, therapy, immune checkpoint inhibitors, immunocompetent animals, syngeneic

## Abstract

Syngeneic in vivo tumor models are valuable for the development and investigation of immune-modulating anti-cancer drugs. In the present study, we established a novel syngeneic in vivo model type named mouse-derived isografts (MDIs). Spontaneous MDIs (sMDIs) were obtained during a long-term observation period (more than one to two years) of naïve and untreated animals of various mouse strains (C3H/HeJ, CBA/J, DBA/2N, BALB/c, and C57BL/6N). Primary tumors or suspicious tissues were assessed macroscopically and re-transplanted in a PDX-like manner as small tumor pieces into sex-matched syngeneic animals. Nine outgrowing primary tumors were histologically characterized either as adenocarcinomas, histiocytic carcinomas, or lymphomas. Growth of the tumor pieces after re-transplantation displayed model heterogeneity. The adenocarcinoma sMDI model JA-0009 was further characterized by flow cytometry, RNA-sequencing, and efficacy studies. M2 macrophages were found to be the main tumor infiltrating leukocyte population, whereas only a few T cells were observed. JA-0009 showed limited sensitivity when treated with antibodies against inhibitory checkpoint molecules (anti-mPD-1 and anti-mCTLA-4), but high sensitivity to gemcitabine treatment. The generated sMDI are spontaneously occurring tumors of low passage number, propagated as tissue pieces in mice without any tissue culturing, and thus conserving the original tumor characteristics and intratumoral immune cell populations.

## 1. Introduction

In the late 1990s and the early 2000s, the blockade of inhibitory immune checkpoint molecules was found to result in the rejection of established tumors and to induce immunity in the secondary exposure to tumor cells [1,2,3,4,5]. These studies led to the stronger focus of experimental studies on the drug development of immune-modulating drugs with syngeneic tumor models in immunocompetent animals. However, the availability of such models is limited to the commonly used murine and cell-line based syngeneic standard tumor models [5].

The establishment of inbred mouse strains in the 1930s, 1940s, and 1950s was an important step in studying the various causes and mechanisms of tumor development as well as new therapeutic approaches in syngeneic animal models [6,7,8]. These can be divided into three groups: subcutaneous or orthotopic grafts of tumor material into syngeneic animals, physically (e.g., ultraviolet (UV) light) or carcinogen-induced primary tumors, and genetically engineered mice (GEM) reflecting specific cancer genotypes known to be involved in tumor development or therapeutic intervention in patients [9,10,11]. These models mainly focused on the interaction between the immune system and the developing tumors [6,7,12]. In parallel, these animal models have been used to determine efficacies and toxicities of novel anti-cancer agents before entering clinical trials. Unfortunately, preclinical drug development was often followed by failure in clinical trials [13,14].

In the 1970s, the possibility of transplanting human tumor material into immunodeficient mouse strains [15,16] raised the hope of enhancing the predictability and reproducibility of preclinical results in patients [17,18]. This hope was only partially fulfilled [19]. One important reason for this is the cell line-based approach. The patient’s primary tumor tissue was disrupted enzymatically or mechanically and the cells were adapted to grow in artificial tissue culture media. Cells often undergo a process of developing genetic alterations [20]. Based on the selection of in vitro cell survival, cell cultures become phenotypically homogeneous with tumor resident stroma cells or proteins interacting with the cancer cells being eliminated [21].

To overcome some of these limitations, in the 1990s, another type of human tumor engraftment was created: patient-derived xenografts (PDXs). Here, solid patient-derived tumor pieces are implanted into immunodeficient mice without prior tissue culture, thus conserving the original tumor characteristics [22]. The tumor pieces include parts of the human stroma [23,24,25,26] and are therefore more physiologically relevant [22,27,28]. PDX tumor cells show lower genetic alterations after in vivo passaging in mice over multiple generations compared with in vitro cell culturing [29,30,31]. Hence, the implanted tumor tissues generally better maintain the genetic and epigenetic abnormalities found in patients [32]. A remaining limitation of PDX models is the growth of the tumor pieces solely in immunodeficient animals [33], which disables the study of immune-modulating drugs or anti-cancer agents interacting with the immune system [22,29]. Currently, this limitation is overcome by the reconstitution of a functional human immune system in animals, e.g., NOD/SCID or NSG mice [33,34,35], through the generation of humanized mice [36,37,38]. Immune or hematopoietic stem cells of various healthy donors, often accompanied by adding fetal thymus or liver cells, are being used for implantation to mimic natural variations in the immune response in patients. However, the high variability and cost renders this model unsuitable for efficacy testing during the early drug discovery phase [39]. Whether the reconstituted, often allogeneic, immune system behaves the same as in the autologous tumor patient has to be addressed [37,40]. The implanted human immune cells may become hyper-activated due to exposure to xenogeneic mouse tissues in a similar fashion as during the alloreactive graft versus host disease [41].

The goal of our present study was to create a cancer model that combines the positive aspects of the current models: the functional immune system of syngeneic mice and the conserved tumor characteristics as well as the wide tumor diversity of PDX. Although mouse and human immune systems may differ in certain aspects [42,43,44,45,46], they display many common and phylogenetically conserved cellular and molecular mechanisms [38,43,44,46]. Therefore, syngeneic models are valuable for the investigation of immunotherapeutic drugs [5,47,48]. This includes the creation of various tumor entities in genetically diverse mouse strains [49,50,51,52,53], for example, in C57BL/6N or BALB/c, prototypically Th1- and Th2-type mouse strains [54,55,56,57,58]. We established a total of 17 new tumor models that were derived from nine spontaneous and eight carcinogen-induced tumors [59] of various immunocompetent mouse strains. The tumors were propagated as tissue pieces in mice only in a PDX-like manner without any in vitro tissue culturing, which guarantees mostly conserved tumor characteristics and intratumoral immune cell populations. These models were named mouse-derived isografts (MDIs) and are suitable for studying the interactions between the stroma, immune cells, and tumor environment in cancer progression, metastasis, and therapy in a completely natural host. The MDI models include adenocarcinomas, carcinomas, sarcomas, or lymphomas.

Here, we outline the establishment and characterization of the spontaneous MDIs and assess their implications for future research. Carcinogen-induced MDIs are introduced in the accompanying paper [59].

## 2. Results

### 2.1. sMDI Establishment History

For the establishment of spontaneous mouse-derived isografts (sMDIs), 36 naïve and untreated animals were monitored for more than two years (Figure 1). During the entire observation period, six mice (two female C57Bl/6N, two male C57BL/6N, and two female DBA/2N) were found dead for unknown reasons. Four mice had to be euthanized for ethical reasons other than tumor signs (three with strong skin irritations and one with signs of paralysis). In two (female C3H/HeJ and female albino C57Bl/6N), suspicious tissues (uterus, abdominal cyst, pancreas, diaphragm, or liver) were found and re-transplanted. Tumor growth was only observed for the diaphragm tissue transplants of female C57Bl/6N albino mouse JA-0013 (Figure 2).

In the other mice, the first tumorous growth was detected after about one year in the female C3H/HeJ mouse JA-0021, which was euthanized for ethical reasons (bad general conditions). Starting from various suspicious tissues (not shown), an enlarged lymph node derived from a female C3H/HeJ mouse was re-transplanted. The transplanted lymph node tissue showed only weak growth in the immunodeficient SCID/bg mouse 1205-16 with a steady state size of 130 mm^3^ at day 177. At necropsy, an enlarged thymus was found in this mouse. Re-transplanting the SCID/bg thymus tissue led to subcutaneous growth in SCID/bg mice, but not in the C3H/HeJ mice. Re-transplanted primary C3H/HeJ lymph node tissue did not grow anymore and was histologically assessed as a questionable lymphoma or even non-malignant morphology (see below and Appendix A). Consequently, this tumor most probably originated from the re-transplanted thymus tissue of the first SCID/bg recipient.

After about one and a half years, palpable tumors in the left and right region of the axillary lymph nodes were detected in the female DBA/2N mouse JA-0009 (Figure 3). During necropsy, splenomegaly was also observed in this animal. Transplantation of the tumor tissues led to stable outgrowth between days 21 and 23 in syngeneic mice.

Simultaneously, a slowly growing tumor was detected in the left hind leg in the female CBA/J mouse JA-0017. At necropsy, the tumor was completely encapsulated and was about 2 cm in diameter (Figure 3). Frozen tumor pieces displayed stable growth in 12 recipient mice between days 81 and 158 (two animals died for unknown reasons at days 53 and 74).

About one month later, the female DBA/2N mouse JA-0011 had to be sacrificed due to ethical reasons. During necropsy, a suspicious and enlarged liver and kidney as well as a gut tumor were detected (Figure 2). The growth of re-transplanted JA-0011 tissue pieces (ranging from 27 to 72 days) was very aggressive. The gut and kidney tissue showed stable growth. However, in some cases, the transplanted tissue(s) did not grow subcutaneously but massively invaded the spleen and/or liver of the recipient mice. Sometimes, this led to unexpected death without any signs of previous heavy tumor burden in the animals. The invaded spleens and livers showed similar histopathology to the primary and subcutaneous tumors (Figure 2). These tissues were successfully re-transplanted.

Another month later, the male C57BL/6N albino mouse JA-0034 had to be sacrificed due to ethical reasons. Different suspicious tissues (gut, liver, lymph nodes, spleen, and salivary gland tissue) were detected during necropsy (Figure 2). Various re-transplanted tissue pieces of JA-0034 grew in the recipient mice. Gut tissue displayed the most homogeneous growth (about 45 days) and was not pursued any further.

Twenty-three months after starting our observations, a big node was palpated on the back of the female C3H/HeJ mouse JA-0023. During necropsy, the tumor was found to be a blood-filled cyst (Figure 3). The tissue was re-transplanted and grew between days 29 and 45 with moderate final tumor volumes of less than 1000 mm^3^ and again displayed a blood-filled, cyst-like phenotype in three out of seven recipient animals. In parallel, a solid kidney-enveloping tumor was detected in JA-0023 (Figure 3). After re-transplantation, tumor pieces grew well within 60 days and reached tumor volumes of more than 1000 mm^3^ in the recipient mice.

Due to significant weight loss and general bad condition, a male BALB/c mouse JA-0032 was sacrificed after two years of observation and a necropsy was performed. Both lungs displayed large solid tumors that grew 45–59 days after re-transplantation in syngeneic animals (Figure 3).

Another month later, the female CBA/J mouse JA-0018 displayed noticeable respiratory problems and was terminated. Different suspicious tissues (lung metastases, enlarged lymph nodes, spleen, and genitourinary tract tissue) were detected during necropsy (Figure 2). Various re-transplanted tumor tissues of JA-0018 grew in recipient mice. The most stable outgrowth (about 60 days) was found with lung metastatic tissue, which was not pursued any further.

The mouse strains, sex, tumor types, and general growth characteristics of the established sMDI models as well as the general histopathological diagnoses are summarized in Table 1 and Figure 2 and Figure 3.

During the studies that lasted for more than two years (total observation time was about 28 months), nine sMDI tumors were finally established, i.e., a stable re-transplantation into syngeneic recipients was performed using frozen tumor pieces. Suspicious tissues were detected and the primary isolates were re-transplanted in seven other mice (female BALB/c, female C57BL/6N albino, female C3H/HeJ, male C57Bl/6N, male BALB/c, male BALB/c, and male C57BL/6N albino). The tumor and tissue pieces were frozen until further use. Therefore, in about 16 of the 36 investigated mice, primary spontaneous malignant growth was observed. However, in the latter seven cases a stable outgrowth of potential frozen sMDI samples has to be verified as the basis for further development.

### 2.2. sMDI: Histological and Pathological Analysis

To characterize the pathological phenotypes of sMDIs and to verify their phenotypic stability after several rounds of re-transplantation, a hematoxylin and eosin (H&E) stain was performed on the tissue sections. For this purpose, primary isolates (tumor or suspicious tissue) and subcutaneous follow-up re-transplants were compared. Whether the tumor/suspicious tissues from the primary isolates actually represented the primary tumor or secondary metastatic tissue could not be determined for most cases. Therefore, the organ of origin remains to be determined, for example by immunohistochemistry or other methods, in future studies. Histologically, we differentiated two main groups of sMDI solid tumor models: of hematopoietic or of non-hematopoietic origin (Table 1). Figure 2 summarizes four various lymphoma sMDIs and one histiocytic sarcoma/histiocyte-associated lymphoma (HS/HAL) model, whereas Figure 3 shows four adenocarcinoma (ADC) sMDI (JA-0023, with two different primary localizations) models in various mouse strains. Single models are described below.

#### 2.2.1. Hematopoietic Tumor Models

Tumor JA-0011/and others (Figure 2, Appendix A) was indexed as a histiocytic sarcoma (HS) or histiocyte-associated lymphoma (HAL) in several tissues (liver, gut, reproductive tract, kidney, and spleen), with both histiocytic and lymphocytic components. Re-transplanted subcutaneous tumors 0339-17, 4007-16, or 0057-17 (Figure 2, Appendix A) also invaded the spleens or livers of recipient mice, e.g., 0052-17 or 1298-16, and displayed a similar histopathological outcome, except for case 1298-16 which had features of malignant lymphoma. Hence, in two re-transplanted cases, a phenotypical HAL (1298-16) and a very typical HS (0052-17) with minimal lymphoid component were diagnosed from the primary index HS/HAL case. To determine any actual HS/HAL phenotype drifting, immunohistochemistry has to be performed in future experiments [60]. Tumors showed partially extramedullary hematopoiesis and a significant infiltration of neoplastic cells by polymorphonuclear leukocytes (neutrophils and eosinophils), lymphoid cells, or lymphoma-like appearance with abundant histiocytic and macrophagic recruitment.

The phenotype of tumor JA-0013/0028-17 (Figure 2, Appendix A) was most consistent with malignant lymphoma, which were probably of B-cell lineage with the presence of cells differentiating towards a plasma cell phenotype and presence of few Mott cells containing Russell bodies. Another re-transplanted tumor, 0027-17, had a few uncommon non-neoplastic multinucleate giant cells and syncytial macrophages leading us to suspect an opportunistic infection or alternatively histiocytic differentiation, thus possibly displaying characteristics of a histiocytic lymphoma. Special stains are necessary for a definitive diagnosis of this particular case.

The female CBA/J-derived tumor JA-0018/0113-17 tissue was isolated from the lung, and was probably a secondary localization, since it was characterized as a malignant lymphoma. The same histology was observed in the re-transplanted tumors. Primary in situ location, typical H&E stains of tumor tissues, and primary diagnoses are shown in Figure 2 and Appendix A.

In index case JA-0034/1426-16 (Figure 2, Appendix A) of a lymphoma, minimal phenotypic differences were observed between the primary and secondary tumors. Original and daughter tumors invaded into adjacent and/or adipose tissues. For further lineage definition (B cells versus T cells versus other), characterization by immunohistochemistry has to be used.

In the case of tumor JA-0021/1571-16, re-transplanting suspicious thymus tissue from primary host animal 1205-16 resulted in further growth in SCID/bg mice. Histology represented a malignant lymphoma in the thymus-derived s.c. tumors, which suggests the origin of the sMDI lymphoma named JA-0021 as the SCID/bg mouse 1205-16 thymus and not the C3H/HeJ lymph node which showed a questionable lymphoma or even non-malignant morphology in the skin mammary teat (see Appendix A).

#### 2.2.2. Non-Hematopoietic Tumor Models

Female JA-0009 DBA/2N mice displayed enlarged axillary lymph nodes and splenomegaly. Despite the gross aspects of this index mouse at necropsy suggesting a hematopoietic tumor, an adenocarcinoma (ADC) was diagnosed.

The epithelial origin of these tumors (see also JA-0009/1286-16, Figure 3, Appendix A), was most likely exocrine, such as mammary gland (breast), apocrine sweat, or possibly less likely pancreas, salivary, prostate, or other epithelium, with adenoid or tubular structures such as what could be observed in the lung, bile duct, airways, or kidney. Invasion of more neutrophils than small lymphocytes in the carcinomatous tissue was observed. Both samples showed infiltrative growth, low to moderate stroma host response, coagulative and disintegrative but overall minimal grade necrosis, and occasional abortive mitosis (mito-necrosis) with abnormal spindle (mitotic index: 3.2). There were areas of neoplasia, suggesting an epithelium to mesenchymal transition and cytoplasmic eosinophilia possibly being epidermoid differentiation, which would have to be confirmed by immunohistochemistry.

Index case JA-0017/0096-17 (Figure 3, Appendix A) an ADC seemed upon first analysis to exhibit some significant degree of phenotypic drifting. Tumors derived from the two daughter mice (1234-16 and 0096-17) were morphologically different. However, the primary index complex type ADC was a mixed cell tumor (of glandular and myoepithelial components) with two neoplastic differentiations. Thus, the re-transplanted tumors might have had various degrees of each of the two components, which could be further characterized using immunohistochemistry.

Different histological diagnoses of index case JA-0023 (Figure 3, Appendix A) were obtained for growing primary tumors with either mammary invasive ADC in JA-0023 (back tumor) and a possible clear cell-like adenocarcinoma (CCA) in JA-0023 (kidney tumor) mimicking renal clear cell carcinoma in humans (Appendix A). The re-transplanted tumors of the kidney sMDI, e.g., 0001-17, had a morphology very similar to the mammary index tumor of the back JA-0023 as well as its daughter sMDI 0005-17 (Appendix A). Therefore, the tumor of origin was likely a mammary ADC. The primary JA-0023 back tumor may mimic a CCA by invading the kidney tissue, or alternatively are 2 different tumors of 2 distinct cell origin (renal tubular cells vs. mammary gland cell). Nevertheless, the index subcutaneous tumor, which was invading vessels and developed from mammary gland and invading into the sub-cutaneous adipose tissue with intense lymphoid infiltration, was also characterized by a lineage with invasive properties and vascular embolization and mild to moderate peripheral mononuclear cell infiltration, mostly by lymphoid cells.

One exception regarding the conserved original tumor characteristics as well as intratumoral immune cell populations after re-transplantation seemed to be tumors JA-0032 and 0055-17 (Figure 3, Appendix A). The original sample was identified as a typical bronchioloalveolar papillary adenocarcinoma invasive in bronchus associated lymphoid tissue, whereas the daughter tumor was a solid trabecular adenocarcinoma with clear cells (cuboidal jointed cells with clear cytoplasm in 0055-17) with small areas of columnar cells, exactly like the index case. The primary proliferative alveolar tissue of JA-0032 invaded the nearby main bronchus. The alveolar walls were obscured by the proliferative tissue. The delicate stroma induced by the neoplastic tissue was variably thickened by collagenous tissue and infiltrated by moderate numbers of small mononuclear cells. In contrast, a solid and very fragile (due to nearly absent supportive stroma and secondary tearing and hemorrhage) proliferation of mid-sized polyhedral to cuboidal epithelial cells was observed in the 0055-17 sMDI with a characteristically clear (rarefied) cytoplasm. Anisopoikilocytosis and nuclear variability was slight to moderate. These differences have to be further analyzed and confirmed by immunohistochemistry, but a likely explanation is type 2 pneumocyte differentiation with surfactant in a heterotopic subcutaneous localization.

More detailed histopathological characterization, as well as larger microphotographs of hematopoietic as well as ADC sMDI models, are shown in Appendix A. Preliminary assessment of the histological analysis of the host stroma reaction and the inflammatory response including the tumor infiltrating leukocyte index (Appendix A) of some of the sMDI models indicated that sMDI tumor models might be categorized into the recently defined “inflamed” and “noninflamed” tumor immune infiltrate phenotypes [5,61,62]. This was confirmed and refined by flow cytometry and RNA-Seq analysis of sMDI JA-0009.

### 2.3. sMDI: Take Rates and Growth Curves

After the model was established, the growth curves and take rates of frozen tumor pieces in four ADC sMDI models (JA-0009, JA-0017, JA-0023 (back), and JA-0032) were determined (Table 1, Figure 4). In contrast, the probable growth times and take rates of the other five established hematopoietic models were calculated from subsequent experiments of various frozen or directly re-transplanted tumor pieces (Table 1). In order to also use the hematopoietic models in efficacy studies, first, their exact growth curves and take rates were determined.

The growth curves of the four ADC models, JA-0009, JA-0017, JA-0023 (back), and JA-0032, showed very variable individual tumor growth periods within 21–23, 81–158, 29–45, and 45–59 days, respectively, and take rates varying from 66 to 100% (Table 1, Figure 4). *Appearance* (*AP*), i.e., the earliest time points allowing robust randomization at mean tumor volumes between 40 and 150 mm^3^ in the 12 (or less, depending on the take rate) animals varied from 6–7, 49–56, 13–20, and 31–41 days, respectively (Table 1, Figure 4). The resulting observation period and *treatment time window* (*TTW*), were not simply the difference between *appearance* and the determined tumor growth period. Since a varying quantity of animals in the single tumor models had to be sacrificed due to fast tumor growth or ulcerations (ethics), or were found dead for unknown reasons, the number of alive animals had already critically decreased before the end of the growth periods. Therefore, we defined *real running time* (*RRT*) of the models. *RRT* determines the time difference from implantation to the time point when the remaining animal number reaches ~60% of the starting animal group size. Therefore, *RRT* defines *TTW*, i.e., the maximal time range to treat animals after randomization (*AP*) to the potential study end. This allowed us to calculate the realistic study length as well as the necessary group size for statistically reliable analysis for testing, such as the immune checkpoint inhibitors. For the four ADC sMDI models, we outlined potential study endpoints, with *RRT*s (and actual *TTW*s) of about 21–23 (15–16), 81 (25–31), 29 (9–16), or 45 (4–14) days (Table 1).

The models were characterized by variable growth periods and variable growth properties. Heterogeneous growth was impressive in the case of the two slow growing models, JA-0017 (take rate 75%) and JA-0032 (take rate 66%). In both models, some tumors appeared to be slow growing while others grew fast or moderate. It remains unclear if these properties reflect differences between re-transplanted tumor pieces, between individual mice, or if they are caused by chance. The heterogeneous tumor growth allowed us to include at least 6–9 mice from 12 animals after randomization to study the efficacy of anti-tumor treatments in both sMDIs. Models JA-0009 and JA-0023 (back) grew within 21–23 and 29–45 days, respectively, and were more homogeneous (Figure 4). Thus, sMDI JA-0009 was chosen for further detailed model analysis by flow cytometry, RNA-Seq, and efficacy studies.

### 2.4. sMDI: Flow Cytometric Analysis

The material of re-transplanted and outgrown JA-0009 was isolated and processed for flow cytometry analysis. Obtained single cell suspensions were stained with master antibody mixes for T cell panel, myeloid cell panel, and macrophage cell panel. Results of the flow cytometry analysis are shown in Figure 5A.

A quantitative comparison of the immune cell subpopulations in sMDI JA-0009 and the established cell line-based syngeneic standard in vivo tumor models (MC38-CEA, CT26.WT, LL/2, Clone M3, 4T1, RENCA, B16.F10) is shown in Figure 5B. The most striking difference between the newly established sMDI JA-0009 and all other formerly tested syngeneic standard models was a massive infiltration of M2 macrophages and a low T cell number, especially of CD8^+^ T cells.

### 2.5. sMDI/cMDI: Preliminary Results of RNA-Seq

RNA-isolation and RNA-Seq were performed by StarSeq (Mainz, Germany) as whole transcriptome shotgun sequencing analysis (Materials and Methods) of two samples each of sMDI JA-0009 as well as of cMDI JA-2011 and JA-2042, two carcinogen-induced sarcomas [59]. RNA-Seq analysis was mainly conducted to obtain an overview of the target expression for potential new drug development and to identify potential mutations.

As mentioned in Section 2.2, the definite tissue origin of the most outgrowing MDI tumor models could not be identified. Therefore, the gene expression pattern determined by complete RNA-Seq whole transcriptome shotgun sequencing analysis will also provide an additional tool—together with the histological data—to better localize the tumor tissue origin, e.g., by a comparison with the gene expression in different tissues of normal mice available from the mouse ENCODE transcriptome data (METD) database [63] or immunohistochemistry. However, this will be the subject of future investigation. Missing identification of the tumor origin complicated the search for tumor-specific changes related to the respective initial normal tissues.

Therefore, as a first step, the expression profiles of various genes of only three gene families related to tumor malignancy or anti-tumor immune response were created: tyrosine kinase receptors, immune population markers, and an IFN-γ signature, which was composed of multiple interferon-responsive genes involved in innate and adaptive immune activities. The gene expression profiles displayed striking differences in the absolute as well as relative expression patterns ≥500-fold for the various genes of one gene family (Appendix A) within the individual tumor model (sMDI ADC JA-0009 and the two cMDI sarcomas, JA-2011 and JA-2042).

A direct comparison of gene expression between the individual MDI models was not possible since the FPKM (fragments per kilobase million) values were not determined in simultaneous experiments. However, comparing the individual gene expression related to one common internal, low expression reference gene (Appendix A) showed different expression patterns within each MDI, not only when comparing sMDI ADC JA-0009 and the two cMDI sarcomas JA-2011 and JA-2042, but also between the cMDIs themselves (Appendix A). The IRE (internal relative expression) gives one an impression on the strength and variabilities of single gene expression within one gene family of the individual MDI models. This indicates that the relative expression of single genes is different in the individual MDI tumor models; e.g., with an overexpression of Met, CD44, or Ly6C in sMDI JA-0009, of Irf1 in cMDI JA-2011, or Cd4, Axl and Ccl5 in cMDI JA-2042 (Appendix A, and inserts). However, whether these differences are model-, tissue-, or mouse strain-specific remains unclear at the moment.

Therefore, RNA-Seq data further supported or confirmed findings already obtained by other methods. For example, the high CD44 gene expression (mean FPKM: 186.078) of the immune population marker gene family in sMDI JA-0009 (Appendix A) well agreed with the strong intratumoral invasion of M2 macrophages found by the flow cytometry analysis (Figure 5B). The high expression of Ly6C in the same gene family supports the histological finding of the possible mammary origin of this tumor (METD database [63]). Similarly, CD4 expression was found to be enhanced in cMDI JA-2042 (Inserts Appendix A), confirming the flow cytometry data showing a strong CD4^+^ T cell infiltration in this cMDI (Figure 4 in Beshay et al. [59]). This relatively high CD4^+^ T cell infiltration might also be responsible for the moderate effects regarding the immune checkpoint inhibitors observed for JA-2042, and displayed a significant tumor reduction by a combination antibody treatment [59]. In contrast, the data do not provide any evidence what might cause the resistance to antibody treatment e.g., in ICPI non-responder JA-2011.

The preliminary data confirmed that RNA-Seq could generally be used to address various questions regarding MDI tumor models. A more detailed characterization of the gene expression of the whole transcriptome shotgun sequencing analysis of further gene families and in further sMDI and cMDI tumor models will be the focus of subsequent investigations. This may also help to determine the tissue of origin and further characterize the various MDI tumor models.

### 2.6. sMDI: Efficacy Studies

JA-0009 was used in two independent efficacy studies. In the first study, JA-0009 tumors were treated with antibodies against the inhibitory checkpoint molecule (mPD-1) or against the chemotherapeutic gemcitabine (Figure 6A). After randomization at day seven, treatment was performed three times with gemcitabine (300 mg/kg) or with anti-mPD-1 antibodies (10 mg/kg). The study was terminated at day 22 due to a critical tumor burden in the vehicle group (tumor volumes in four animals were >2000 mm^3^). Gemcitabine treatment resulted in significant tumor growth inhibition (*p* = 0.0012) with nearly complete tumor regression in all mice on day 22; anti-mPD-1 treatment showed only a moderate effect that was statistically not significant (*p* = 0.0654).

In the second study, we investigated whether anti-mCTLA-4 treatment showed similar inhibitory effects when compared to anti-mPD-1 treatment, as well as if the anti-mPD-1 effect could be amplified through a combination with anti-mCTLA-4 treatment (Figure 6B). Anti-mPD-1 treatment again resulted in moderate but not significant tumor growth inhibition (*p* = 0.1899). The trend observed in the first study was confirmed. Treatment with anti-mCTLA-4 antibodies resulted in marginal tumor growth inhibition (*p* = 0.3965). The combination of both antibodies did not result in additive effects and showed similar weak effects on tumor growth (*p* = 0.3605) as those shown for the anti-mCTLA-4 antibody treatment alone.

## 3. Discussion

Mouse in vivo tumor models are important tools for studying the various causes and molecular mechanisms of tumor development as well as for investigating new therapeutic approaches [5,10,12,13,47,48]. Limitations of the current mouse models include the differences between the structure and function of the human and mouse immune system and physiology [42,43,44,45,46]. Therefore, these variances must be considered when designing and performing in vivo experiments with the goal of clinical translation [46]. The immunoglobulin-superfamily-based adaptive and innate immune system of humans, mice, and other vertebrates has co-evolved in common ancestors [42]. Therefore, the general structures of the immune system in humans and mice are phylogenetically sufficiently closely related, and also display many common, conserved cellular and molecular mechanisms that enable the use of syngeneic mouse tumors in immunocompetent mice as relevant experimental in vivo cancer models [38,42,43,44,46]. Further selection and studies on relevant syngeneic mouse tumor models are valuable for characterizing new tumor therapeutic and immunotherapeutic approaches [5,47,48].

Here, we established a novel type of spontaneously appearing syngeneic primary tumors from naïve and untreated animals with low passage number that were propagated as tissue pieces in mice only without any prior or subsequent in vitro manipulation, showing mostly conserved original tumor characteristics and intratumoral immune cell populations (Figure 2 and Figure 3, Appendix A). The use of tumor tissue pieces to establish or amplify tumor growth is a valuable procedure in patient-derived xenografting (PDX) and has also been used as an intermediate step in transplanting syngeneic or semi-allogeneic mouse tumors [22,23,24,25,26,27,28,64,65,66]. In the syngeneic mouse situation, this procedure was commonly used as an auxiliary means to establish tumor cell lines from primary tumor pieces either by in vivo manipulation (e.g., intraperitoneal application) or in vitro artificial tissue culture of enzymatic or mechanical disrupted tumor tissues. Cell line-based mouse models have been preferred for a long time, e.g., with Ehrlich Ascites or L1210 cells, since they are good to handle [67,68]. The manipulated cells often show enhanced tumorigenicity when compared to the original tissues [66], and such tumor cell lines additionally allow for the possibility to be used for in vitro investigations of tumor properties, behavior, and therapeutic sensitivity [69].

To the best of our knowledge, we used this technique for the first time to create a novel and separate syngeneic in vivo model type. The primary spontaneous mouse-derived-isograft (sMDI) in vivo cancer models (Table 1) displayed new model quality and properties not available with standard syngeneic or xenograft tumor models. One main advantage of this model is that it is very close to the real clinical situation in patients, representing actual outgrowing spontaneous primary tumors or metastases that have overcome the body’s own regulatory mechanisms. The models are transplantable; they are tumorigenic not only in the primary tumor-bearing animal, but also in other syngeneic fully immunocompetent hosts without any prior or subsequent additional in vivo or in vitro manipulation (Figure 1). We expected that, depending on the mouse strain, sex, or age, about 15–100% of the animals would develop spontaneous solid tumors of non-hematopoietic and hematopoietic origin of different organogenesis at the age of one to two years [70,71,72,73,74]. Only in the case of female C3H/HeJ was a preference of mammary carcinoma or hepatoma anticipated [70]. Finally, in this first study, we were able to establish nine different sMDI tumor models in five different inbred strains, with different sexes (7 female and 2 male) and in three different H-2 MHC class I haplotypes (Table 1, Figure 1 and Figure 2), enhancing the number of available syngeneic tumor models. The apparent dominance of female sMDIs was not confirmed by further outgrowing tumor/suspicious tissue isolates in seven other mice with a female to male ratio of three to four.

In most cases, we could not differentiate whether tumor or suspicious tissues from primary isolates actually represented the primary tumor or the secondary metastatic tissues. Tumorous tissue derived from the lung of JA-0018 CBA/J (lymphoma) or the enlarged axillary lymph nodes and splenomegaly found in JA-0009 DBA/2N mice (adenocarcinoma) are exemplary illustrations of this issue (Figure 2 and Figure 3). In the case of sMDI JA-0021, the mouse strain of the tumor origin also remained unclear for a time, since it was outgrowing from secondary SCID/bg thymus tissue and only growing in immune-deficient mice. As the primary C3H/HeJ lymph node itself showed a questionable lymphoma or even non-malignant morphology (see Appendix A), it is likely that we established by chance a primary SCID/bg lymphoma/thymoma (Figure 2), as already described in NOD-SCID mice [75]. However, whether the lymphoma also grows in H-2^d^ × H-2^b^ immunocompetent F1 hybrids has to be verified.

In summary, during these studies that lasted for more than two years, nine sMDI tumors were finally established, and suspicious tissues from seven other mice were isolated. Therefore, a total of 16 out of 36 (i.e., 16/26 alive, long-term monitored) animals developed primary transplantable spontaneous tumors within the more than two years of observation. This indicates that this method seems to be a valuable tool for the development of novel tumor models to enhance model variety.

The established tumors showing intense vascularization or invasive growth in some models displayed the histopathological properties of solid tumors (Appendix A) [76] such as adenocarcinomas (*n =* 4), lymphomas (*n =* 4), or histiocytic sarcomas or histiocyte-associated lymphomas, HS/HAL (*n =* 1) (Figure 2 and Figure 3, Appendix A). The tumors showed various host stromal or inflammatory reactions and varying numbers of tumor infiltrating leukocytes, which seem to be similar when comparing the primary and re-transplanted tumor pieces (Figure 2 and Figure 3, and Appendix A). First, the flow cytometric analyses confirmed and refined the histopathological findings of tumor infiltrating leukocytes (Figure 5A,B). With regard to the host stroma reaction and inflammatory responses (Appendix A), sMDI tumor models also displayed the defined inflamed or noninflamed tumor immune infiltrate phenotypes [5,61,62]. Therefore, the sMDI models seem to reflect the classical clinical situation in patients as well as heterogeneous tumor types, individual immunological response pattern, and varying phenotypes of the tumor immune infiltrates.

As in patients [77], a histologically heterogeneous intratumoral composition of malignant cells appeared in some sMDIs, e.g., in cases JA-0017, or JA-0032 (Figure 3). The trabecular epithelial carcinoma cells found in one secondary JA-0017 tumor (0096-17, P2F) might have resulted from the actual differentiation of the primary complex mixed adenocarcinoma cells of an unidentified exocrine gland. Whether multiple re-transplantation itself (passage 2) or of frozen material triggers the differentiation remains to be verified since second re-transplantation (1234-16, P1) showed an identical complex mixed histology with primary ADC. The histologically indicated instability of the tumor, i.e., losing the balance of the complex mixed primary ADC, which results in the possible preference of one or another tumor cell population in the follow-up daughter tumors. Whether this imbalance in JA-0017 tumor heterogeneity is able to explain the heterogeneous growth remains to be verified (Figure 4). sMDI JA-0032, a primary papillary lung adenocarcinoma composed of tubule and papillary proliferations, and its secondary (0055-17, P1) tumor, in this case not derived from previously frozen material, also displayed histological heterogeneous phenotypes and heterogeneous tumor growth (Figure 4). Therefore, it must be further verified if growth and histological heterogeneity are general or particularly changing properties of some sMDIs. In addition, whether homogeneous follow-up daughter tumors originating from heterogeneous primary tumors could be established as possibly stable sMDI subtypes should be determined. Regarding the histological variability, whether the heterogeneity in some tumors coincides with the changing numbers and compositions of tumor infiltrating leukocytes should be examined (as indicated by histological findings, e.g., in JA-0017).

sMDI JA-0009 is an adenocarcinoma originating from the exocrine gland of either the mammary gland, pancreas, salivary, prostate, or bile duct. The histologically homogeneous and invasive tumor was characterized by the consistent and fast growth of subcutaneous re-transplants after about 22 days (Figure 4). To obtain insights into the mechanisms and interactions of tumor growth, immune system, and physiological background of the novel sMDI, a detailed analysis of the model was performed. This will be the basis to create or analyze new therapeutic (immunological) approaches. Mosely et al. extensively characterized six commonly used syngeneic standard tumor models by array comparative genomic hybridization, whole-exome sequencing, exon microarray analysis, and flow cytometry [5] to enable investigators to select appropriate models to gauge the activity of feasible immunotherapeutic concepts as well as combinations with targeted therapies for the given in vivo situation. We used a similar approach (flow cytometric analyses, chemotherapeutic or immunotherapeutic, immune checkpoint inhibitor intervention, and preliminary gene expression analysis by RNA-Seq) to characterize sMDI model JA-0009. However, the gene expression pattern determined by complete RNA-Seq whole transcriptome shotgun sequencing were here analyzed merely for various genes of only three gene families related to tumor malignancy or anti-tumor immune response: tyrosine kinase receptors, immune population markers, and an IFN-γ signature. Since the definite tissue origin of the most outgrowing MDI tumor models could not be identified (Section 2.2) a more detailed characterization of the gene expression of the whole transcriptome shotgun sequencing analysis of further gene families and in further sMDI and cMDI tumor models will be the focus of subsequent investigations.

As expected from previous ADC xenograft models and clinical data [78,79,80,81], chemotherapeutic intervention with gemcitabine completely inhibited the growth of primary mouse malignant epithelial tissue of exocrine gland origin (Figure 6A,B). In the future, such complete regression could allow the establishment of therapy-resistant MDI sublines. We established such a subline from a primarily regressed and further relapsed PD-1-sensitive cell line-derived MC38-CEA tumor (Appendix A), which not only changed the antigenic profile (loss of CEA expression), but also became completely PD-1-resistant (Appendix A). In contrast to gemcitabine, JA-0009 was completely insensitive to immune checkpoint inhibitor intervention by PD-1 or CTLA-4 antibodies (Figure 6). Since several syngeneic standard models were also completely insensitive to immune checkpoint inhibitor intervention (Appendix A) [82], we compared the JA-0009 flow cytometry data with the respective results from our syngeneic standard tumor models (colon carcinoma MC38-CEA, colon carcinoma CT26.WT, lung carcinoma LL-2, melanoma Clone M3, breast carcinoma 4T1, renal carcinoma RENCA, and melanoma B16.F10) to determine possible similarities. As expected, anti-immune checkpoint inhibitor treatment of sensitive syngeneic tumors (Appendix A), such as CT26.WT or MC38-CEA, showed a completely different immune cell composition (Figure 5B). However, in the insensitive models (e.g., LL/2, 4T1 or RENCA), the composition of tumor infiltrating leukocytes was also completely different to sMDI JA-0009 (Figure 5B). Histologically, the JA-0009 tumor showed a different situation with few infiltrates of small lymphocytes and neutrophils, with more neutrophils than small lymphocytes in carcinomatous tissue (Appendix A). Flow cytometric analyses confirmed the weak infiltration with CD8^+^ or CD4^+^ T cells, but showed a massive invasion of M2 macrophages into the JA-0009 tumor (>70% of tumor infiltrating leukocytes) (Figure 5B) [5,82]. Since M2 macrophages can act as anti-inflammatories due to the production of immunosuppressive factors such as IL-10, transforming growth factor β (TGFβ), or prostaglandin E2 (PGE2), and the recruitment of immunosuppressive regulatory T cells [83], immune checkpoint inhibitor treatment only moderately affected the tumor growth of JA-0009. Therefore, a tumor promoting function by M2 macrophages can be assumed [84,85], which has been reported in some patients [86,87,88]. Therefore, further studies should try to influence the polarization and differentiation of M2 macrophages in the JA-0009 tumor using antibiotics, shifting gut dysbiosis, or by coumarins [84,89], to verify the actual role of M2 macrophages in this model. Enhanced expression of CD44, CD45, Ly6C1, Granzyme, CXCL10, and CCR5 as confirmed by the RNA-Seq analyses demonstrated the massive macrophage infiltration (Appendix A). However, e.g., abnormal c-Met activation in the epithelial tumor cells (Appendix A) indicates that additional factors might be involved in the poor immunotherapeutic prognosis in sMDI JA-0009 [90].

As in the case of JA-0009, after more detailed histological, immunohistochemical, flow cytometric, genetic, and functional characterization, it will be applicable to determine the individual intratumoral invasion and immunological parameters. This would allow the characterization of therapeutic interventions related to the respective tumor entities, tumor infiltrating leukocyte composition, gene expression, and general immunity stage in the spleen, lymph node, or blood, and enables the selection of suitable models to evaluate the activity of new therapeutic concepts. The four sMDI ADCs derived from three female and one male of four different mouse strains as well, as the four sMDI lymphomas derived from three females and one male of three mouse strains, already provide the possibility of comparing the various characteristics of one tumor type and to test their sensitivity in various mouse strains. The four sMDI ADCs will also allow a preliminary investigation to determine if eventual resemblances or pronounced disparities in the outcome of different therapeutic concepts might reflect strain-dependent, tumor-type (e.g., ADC) related, or individual tumor-specific properties.

However, the currently available sMDIs appear insufficient to answer further questions, e.g., whether tumors (1) of different histopathological classification (e.g., adenocarcinomas, sarcomas, or lymphomas), or (2) of similar histopathology (e.g., various adenocarcinomas) within one single mouse strain display similar or variant genetic, cell biological, or immunological characteristics. Additionally, (3) questions whether these tumors are therefore differentially sensitive or insensitive to certain therapeutic approaches, or (4) if such differences may depend on a distinct MHC class background, remain unanswered, since only one or two tumors per individual mouse strain are accessible, which are even mostly of different histopathological diagnosis.

To address some of these questions, we developed another type of MDI induced by carcinogens, the cMDI [59]. Using carcinogen induction, we established a larger number of individual tumors of a more histologically-related origin in a manageable number of animals and a shorter time frame.

Our newly established MDI models increase the pool of available syngeneic in vivo tumor models, not only in quantity (plus nine), but also in quality (primary, spontaneous, in five to six various mouse strains, few passages, neither in vitro nor in vivo manipulated), and provide the ability to characterize new experimental and translational in vivo cancer therapeutic concepts.

### Future Outlook

The novel sMDI and cMDI models [59] are promising tools for characterizing new therapeutic approaches in the presence of a functional immune system. New possibilities for the syngeneic MDI tumor models have been provided. The inclusion of further inbred mouse strains will enhance the range of models with immunological differences [49,50,51,52], genetic heterogeneity in angiogenesis [91], various susceptibility to certain drugs [92], or strain-specific susceptibility for metastasis [93]. The inclusion of mouse strains with the not-yet-considered phylogenetic H-2 class I and the different class 2 antigenic background, of further carcinogens, or of physical tumor induction, e.g., by ultraviolet (UV)-light or radioactive irradiation, will additionally increase the model range. Finally, it is possible to include histologically stable sMDI subtypes established from homogeneous re-transplanted tumors or therapy-resistant sublines derived from relapsing tumors (Appendix A) after various kinds of treatments (immunological, chemotherapeutic, irradiation or combinations thereof).

## 4. Materials and Methods

### 4.1. Mouse Strains

All in vivo experiments were performed in accordance with the German Animal License Regulations (Tierschutzgesetz) identical to the UKCCCR Guidelines for the Welfare of Animals in Experimental Neoplasia (licenses G-15/99, sMDI, Regierungspräsidium, Freiburg, Germany) [94]. Mice of different strains and sexes (Table 2) were purchased from Charles River Laboratories, Sulzfeld, Germany, at 6 to 14 weeks of age. 

They were kept under pathogen-free conditions in conventional cages stored in Scantainer Ventilated Cabinets (Scanbur, Denmark), holding four animals with autoclaved nesting material, and cardboard tunnels. If the animals exceeded 30 g body weight, they were separated and only 2–3 animals were housed in one cage. Mice were subjected to a 12/12 h light/dark cycle, with ad libitum autoclaved water and a M-Zucht rodent diet (ssniff Spezialdiäten GmbH, Soest, Germany).

### 4.2. sMDI: Establishment and Re-Transplantation

To establish spontaneous tumors from naïve and untreated animals, four mice of each strain, as shown in Table 2, were monitored long-term (≥2 years) for spontaneous tumor appearance or other critical signs indicating spontaneous tumor development (Figure 1). Animal weights were recorded twice weekly (Monday and Friday) during long-term observations and three times weekly (Monday, Wednesday, and Friday) after re-transplantation and during treatment studies. Animal behavior was monitored daily.

Termination criteria: Tumors exceeding 2000 mm^3^ or an edge length of 2 cm, tumor ulceration, weight ≥20%, cachectic phenotype (tumor cachexia), abnormal, non-physiological posture as a sign of pain, apathy (severe inactivity), strongly reduced feed and water intake, severe dyspnea, motor deficit manifestations or paralysis, ascites, persisting diarrhea, massive behavioral changes, or other unexpected signs indicating tumor burden.

Mice were sacrificed according to the ATBW/GV-SOLAS (Arbeitsgemeinschaft der Tierschutzbeauftragten in Baden-Württemberg/Gesellschaft für Versuchstierkunde, Society of Laboratory Animal Science), and all tissues were assessed macroscopically and cut in sterile phosphate-buffered solution (PBS) in petri dishes by scalpels into small pieces (2–3 mm^3^). The tumor pieces were implanted subcutaneously into the flank of 5- to 6-week-old sex-matched syngeneic and/or female immunodeficient SCID/bg mice using a trocar. Implantation was performed under anesthesia with 2–3 volume percent isoflurane in combination with an oxygen flow rate of 0.6 L/min. If possible, tumor pieces were also frozen and stored in 10% dimethyl sulfoxide (DMSO) freezing medium at –80 °C or colder. Outgrowing tumor volumes were determined by caliper measurements two times a week. Tumor sizes were calculated according to the formula W^2^ × L/2 (L = length and W = the perpendicular width of the tumor, L > W). During necropsy, tumor tissue and other suspicious tissues were collected. In the next step, non-necrotic tissue areas were cut into small pieces (2–3 mm^3^), and amplified once again by re-transplantation into syngeneic, sex-matched, and/or female immunodeficient SCID/bg mice. Excess tumor/tissue pieces were frozen stored until further use. Amplification and sample collection were repeated several times, and samples were called F1–F*n* (*n* = max. 5–10 amplifications) [95]. Certain tumor pieces (primary and re-transplanted) were directly fixed in formalin during necropsy for further analysis. The main part of model establishment was the re-transplantation and stable outgrowth of previously frozen and stored tumor pieces into syngeneic animals. To this end, the frozen tumor pieces were thawed at 37 °C and washed twice in ice-cold PBS, and then implanted by trocar as described above.

### 4.3. sMDI: Histological and Pathological Analysis

Wet tissues were collected in embedding cassettes and formalin-fixed in 4% neutral buffered formaldehyde (Engelbrecht, Edermünde, Germany) at room temperature for about 24 h, followed by automatic dehydration and embedding in IHC-grade paraffin using a Leica TP 1020 (Leica Biosystems, Nussloch, Germany) and Leica Histo Core Arcadia H/C (Leica Biosystems). FFPE tissue blocks were stored at room temperature. Sections of the paraffin blocks were cut into slices of approximately 2–3 µm thickness and stained with H&E at TPL Path Labs (Freiburg, Germany).

Histopathological examinations were then performed blind using a Zeiss Axioscope microscope (Carl Zeiss, Jena, Germany) at a magnification of up to 400×, by one of the authors (TL). Digital microphotographs were taken using a Nikon Digital Sight DS-Fi camera (Nikon Instruments Europe B.V., Amsterdam, Netherlands). Whole slide imaging using an Axioscan Z1 (Carl Zeiss) were also performed from all case, for the purpose of comprehensive iconography.

### 4.4. sMDI: Establishment and Efficacy Studies

To assess the growth and take rates of the re-transplanted and histopathologically characterized sMDIs, frozen tumor pieces were re-transplanted by trocar into 12 sex- and strain-matched animals. Primary tumor volumes were determined by caliper measurements. The health status was documented, and animals were euthanized individually before study termination when the above-mentioned ethical termination criteria were reached. The take rate is defined as a percentage of total tumor-bearing/developing versus all re-transplanted animals during the observation period. Tumor growth appearing only outside the observation period was not included in the analysis.

Frozen tumor pieces of the sMDIs were thawed at 37 °C and washed twice in ice-cold PBS, and then implanted by trocar into an appropriate number of sex- and strain- matched animals for efficacy studies. Implantation and health monitoring were performed as previously described in Section 4.2. In the following, the primary tumor volumes were determined by caliper measurements. After the primary tumors reached the sMDI respective mean volume (defining randomization criteria), tumor-bearing animals were randomized into three or four groups of 10 to 12 animals. For randomization, a robust automated random number generation within individual blocks was used (Excel 2016, Microsoft, Redmond, WA 98052, USA). Treatment was initiated on the same day or the day after, according to the following schedules: vehicle control (PBS or isotype control antibodies) or antibodies anti-mPD-1, anti-mCTLA-4, or a combination thereof [96] were injected intraperitoneally three times every third or fourth working day with a dosage of 10 mg/kg and an injection volume of 10 mL/kg or 5 mL/kg. Anti-mPD-1 antibody clone RMP1-14, rat IgG2a, and anti-mCTLA-4 antibody clone 9D9, mouse IgG2a were used. All antibodies for treatment were purchased by BioXCell, Lebanon, NH 03784. Chemotherapeutic drug gemcitabine (Pharmacy, University Medical Center Freiburg, Germany) was injected intravenously every fourth day with a dosage of 300 mg/kg and injection volume of 5 mL/kg [78,79]. Tumor growth and effects regarding tumor growth reduction due to therapeutic intervention were followed by caliper measurements until the end of observation. The study was finalized by necropsy, and the tumor volumes and wet weights were determined. In some cases, parts of tumors from various experimental groups were also formalin-fixed and paraffin embedded and used for flow cytometry analysis and/or for RNA-sequencing. Probability (*P*) was tested with the parametric unpaired *t*-test (GraphPad Prism 5.04) when compared to the PBS vehicle control. Differences were determined as not significant with ns > 0.050 and as significant when *p* < 0.050, *p* < 0.010, and *p* < 0.001.

### 4.5. Flow Cytometric Analysis of sMDI Infiltrating-Immune Cells

At least 300 mg of tumor material was disrupted using gentleMACS™ C Tubes (Miltenyi Biotec, Bergisch Gladbach, Germany) containing the enzyme mix of the Tumor Dissociation Kit (mouse), according to the manufacturer’s instructions. Erythrocytes were removed with the Red Blood Cell Lysis Solution (Miltenyi Biotec). Single cell suspensions were counted and dispensed in 96-well plates if possible at 3 × 10^6^ cells/well (if there were less than 2 × 10^6^ single cells in total in a tumor sample, the T cell panel was preferred for staining). The single cells were washed with PBS (Capricorn, Ebsdorfergrund, Germany) and stained for live cells with eBioscience™ Fixable Viability Dye (Thermo Fisher, Nidderau, Germany). After washing and centrifugation (400 *g*) the samples were incubated with 50 µL/well Fc block (anti-mouse CD16/CD32, 1:50) for 30 min in FACS^TM^ buffer. Thereafter, a 2× concentrated master antibody mix (T cell panel with antibodies against murine CD45, CD3, CD4, CD8a, CD25, CD44, myeloid cell panel with antibodies against murine CD45, CD11b, CD11c, Ly6G, and Ly6C, and MΦ panel CD45, CD11b, F4/80, MHC II, CD206) was added to each well (50 µL) and incubated for 30 min in the dark. After washing, cells stained with the myeloid or macrophage antibody panel were fixed with 2% paraformaldehyde in PBS. Cells stained with the T cell antibody panel were prepared for intracellular staining. Intracellular staining was primed by adding 50 µL fix/perm buffer for 30 min. Thereafter, 1× permeabilization buffer (1:10 diluted with water; 100 µL) (Thermo Fisher Scientific) was added and cells were centrifuged at 836× *g*. The cell pellet was resuspended in 1× permeabilization buffer containing the anti-FoxP3 antibody and incubated for 30 min in the dark. After washing twice with 1× permeabilization buffer, cells were washed with the FACS^TM^ buffer. The samples were analyzed by flow cytometry using a BD LSR Fortessa^TM^ (Becton Dickinson, Heidelberg, Germany). All antibodies were purchased from Thermo Fisher with the exception of CD3 and CD206 (BioLegend, San Diego, CA 92121, USA) and CD8a (Becton Dickinson).

### 4.6. sMDI: RNA-Seq

Total RNA integrity and quantity was analyzed (StarSeq, Mainz, Germany) using the RNA Nano chip assay on an Agilent Bioanalyzer 2100 (Santa Clara, CA 95051, USA) and the Qubit RNA HS assay (Thermo Fisher Scientific). Poly-A+ isolation from total RNA, fragmentation of mRNA, and synthesis of double stranded cDNA was performed according to the NEB Next Ultra II Directional RNA Library Prep Kit for the Illumina protocol (New England Biolabs, Ipswich, UK). End-repaired, A-tailed, and double indexed adaptor-ligated cDNA was polymerase chain reaction (PCR)-amplified by 14 cycles. The libraries with an average size of 450 bp were sequenced in paired-end mode (2 × 151 bp) on an Illumina NextSeq 500 System (50 million paired-end reads of each library). Fragments per kilobase million (FPKM), from the double determination of two independent tumor samples each, were determined and depicted for a number of genes. The values from a single RNA-Seq experiment, i.e., from a single MDI model, thus reflected the objective gene expression pattern within the individual MDI tumor model.

However, FPKM values are not directly comparable across separate RNA-Seq experiments, i.e., across the different independently analyzed MDI tumor models [97]. However, if the majority of analyzed genes in various MDI samples display very similar, either low or high, FPKM patterns (Appendix A), an indirect, careful quantitative comparison of the expression patterns among these tumors is acceptable. Internal Relative Expression (IRE) of RNA-Seq determined FPKM-Values was calculated as quotient comparing individual gene expression within individual MDI models each, related to the expression of an internal, low FPKM-value reference gene (red triangle-Appendix A). Data are shown as n-fold of expression of respective reference gene. For each gene family one common reference gene, Ctla4 (IFN y signature), Cd8b1 (immune population marker), and Flt4 (tyrosine kinase receptor) was selected within the MDI models.

In contrast, a quantitative comparison with publicly available FPKM or RPKM values for different adult tissues of normal mice, for example from the METD database [63], was not performed since a definite characterization of primary tumor origin was not possible in most cases with the present histopathological analysis (Section 2.2).

## 5. Conclusions

The sMDI and carcinogen-induced MDI (cMDI) [59] models are promising new in vivo tools representing primary tumors with conserved tumor characteristics and intratumoral immune cell populations that were histologically and genetically characterized. Spontaneous tumor models do not represent any malignant degeneration, but instead represent outgrowing tumors (or metastases) that have overcome the body´s own regulatory mechanisms, thus corresponding to clinically manifested tumors without any manipulation.

Inclusion of both sexes and various mouse strains with different MHC haplotypes and immunological background increase the therapeutic possibilities. The established sMDI and cMDI models can be further refined. Use of relapsing tumors from original tumors after various kinds of treatments (immunological, chemotherapeutic, irradiation, or combinations thereof) or strains with specific immunologic deficits, such as knockouts and transgenics, could provide further possibilities for finding treatments for such therapy-resistant re-growing tumors. Our models provide a broad experimental range for in vivo cancer therapeutic concepts.

## Figures and Tables

**Figure 1 cancers-11-00244-f001:**
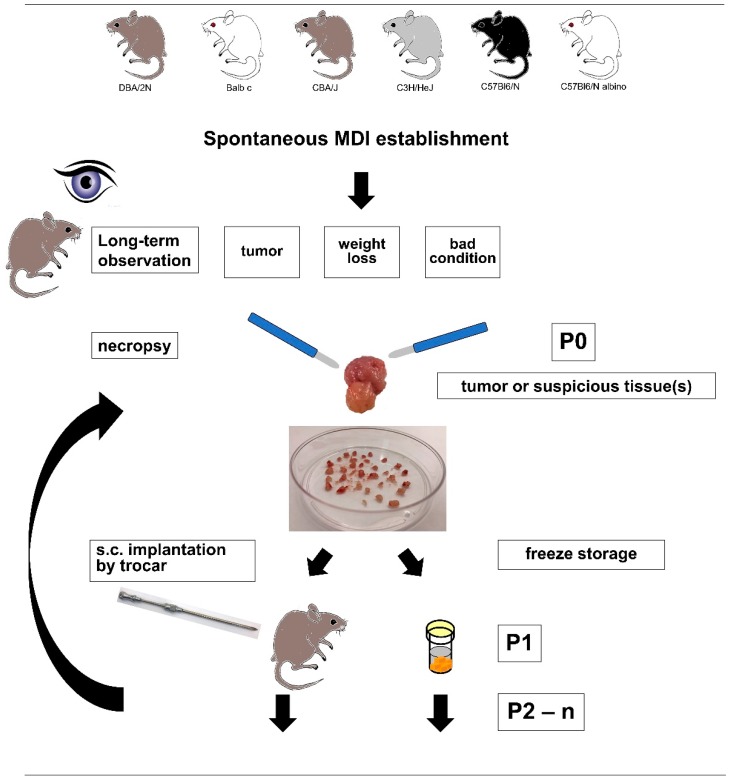
Schematic schedule of spontaneous mouse-derived isografts (sMDI) detection, selection, and establishment. The figure schematically shows the included mouse strains as well as the termination criteria and further handling of the tumor or other suspicious tissue samples. (s.c. = subcutaneous implantation).

**Figure 2 cancers-11-00244-f002:**
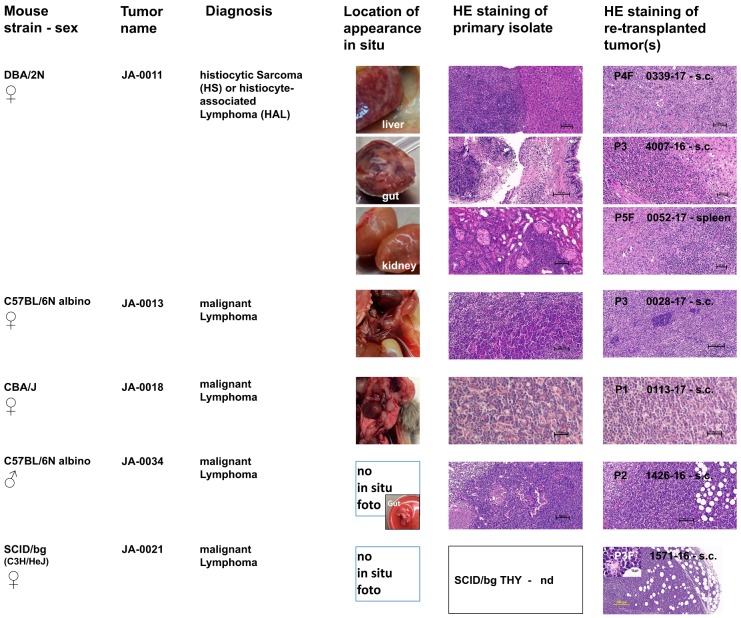
Histopathological characterization of five lymphoma or HAL/HS sMDI. In situ location as well as hematoxylin and eosin (H&E) staining of primary isolated tumor or suspicious tissue and of re-transplanted subcutaneous (s.c.) or spleen tumors (passages P1–P3), or frozen tumor tissue pieces (passages P2F–P5F) were documented by photographs of five hematopoietic sMDIs (JA-0011, JA-0013, JA-0018, JA-0034, and JA-0021). Three tumors were isolated from the gut, liver, or kidney of JA-0011. The histological finding that primary C3H/HeJ isolated lymph node of the sMDI lymphoma named JA-0021 showed a questionable lymphoma or even non-malignant morphology (see Appendix A) indicates that malignant tissue might probably originate from the suspicious, re-transplanted SCID/bg thymus (THY) tissue (HE nd = not done) of the primary recipient H-2^d^ × H-2^b^ SCID/bg mouse 1205-16 and not from the primary C3H/HeJ mouse JA-0021. For more details see: Appendix A.

**Figure 3 cancers-11-00244-f003:**
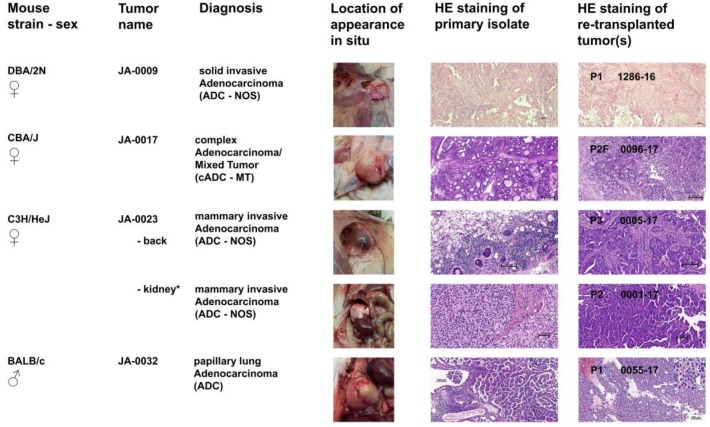
Histopathological characterization of four adenocarcinoma sMDI. In situ location as well as H&E staining of primary isolated tumor or suspicious tissues and of re-transplanted subcutaneous tumors (passages P1–P3), or frozen tumor tissue pieces (passage P2F) were documented by photographs for four selected non-hematopoietic tumor models (JA-0009, JA-0017, JA-0023, and JA-0032). For JA-0023, two malignant primary tissues were isolated from the back and kidney. Original diagnosis of kidney tumor was a clear cell-like renal adenocarcinoma. Re-transplanted kidney tumors (e.g., 0001-17), however, displayed the same histology as the primary back tumor of mammary invasive ADC. For more details see: Appendix A.

**Figure 4 cancers-11-00244-f004:**
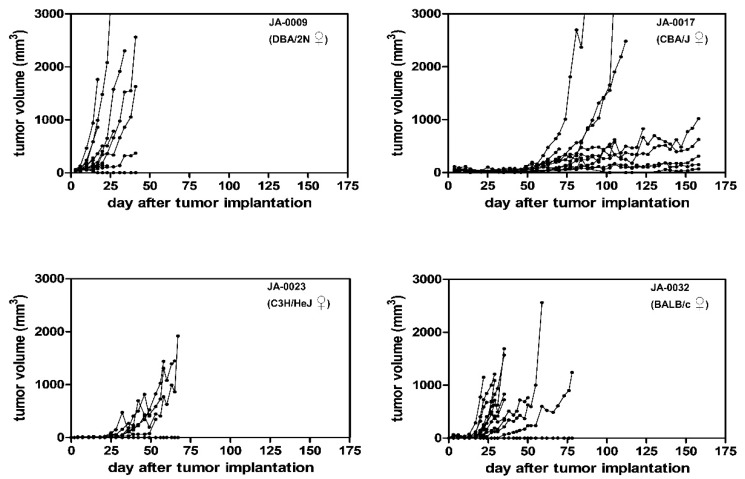
Take rates and growth curves of the selected sMDI models. Growth curves of 12 individual mice each implanted with previously frozen tumor pieces are shown for the sMDI models JA-0009, JA-0017, JA-0032, and JA-0023 (with tissue originating from a primary back tumor) in syngeneic female DBA/2N, CBA/J, BALB/c, and C3H/HeJ mice, respectively. Tumor growth was measured twice weekly with calipers.

**Figure 5 cancers-11-00244-f005:**
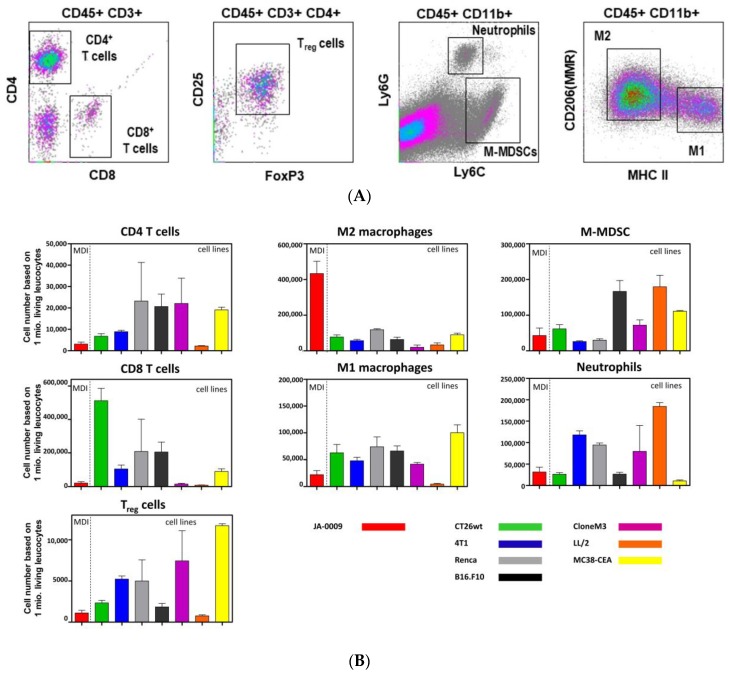
(**A**) Flow cytometric analysis of tumor infiltrating leukocytes (density dot plots) in sMDI JA- 0009. Representative flow cytometry blots of tumor infiltrating CD4^+^/CD8^+^ T-cells, Treg cells, M-MDSCs, neutrophils/G-MDSC (cells of CD11b^+^, Ly6G^+^, Ly6C^int^ phenotype), and M1/M2 macrophage subpopulations isolated from untreated syngeneic adenocarcinoma mouse model JA-0009 sMDI. M-MDSC = monocytic myeloid-derived suppressor cells, MMR = macrophage mannose receptor, Treg = regulatory T cells. (**B**) Quantitative flow cytometric analysis of tumor infiltrating leukocytes in sMDI JA-0009 in comparison to seven cell-line based syngeneic standard tumor models. The flow cytometry analysis of tumor infiltrating leukocytes in sMDI JA-0009 was compared to seven established cell-line based syngeneic mouse standard models for CD4^+^/CD8^+^ T-cells, M-MDSCs, neutrophils, M1/M2 macrophages, and Treg cell subpopulations were isolated from the untreated tumor tissue. For the antibodies and procedure used here, see Materials and Methods. The graphs depict the number of cells of each subpopulation per 1 × 10^6^ living leukocytes. Further information regarding the used cell line-based syngeneic standard tumor models is provided in Appendix A.

**Figure 6 cancers-11-00244-f006:**
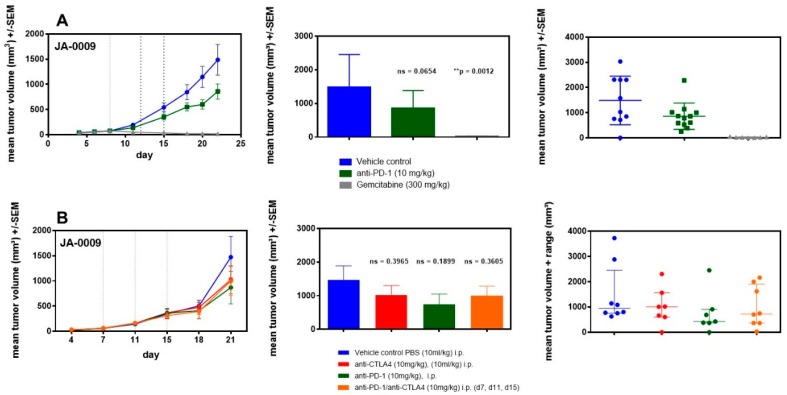
(**A**). Two independent efficacy studies with chemotherapeutic gemcitabine or antibodies against the inhibitory checkpoint molecules mPD-1 and mCTLA-4 in the sMDI JA-0009 tumor model. Efficacy study of the sMDI JA-0009 tumor model characterized the effects of antibodies against the immune checkpoint inhibitor mPD-1 as well as gemcitabine as a chemotherapeutical agent. (**B**). In the second study treatment, the effects of antibodies against the immune checkpoint inhibitor mPD-1, mCTLA-4, or both were tested. Mice were randomized on day 7. Dotted lines show the timepoints of treatment. Results are shown as growth curves (curve chart), mean of groups (bar graphs), and individual values of a single mouse per group at the study termination (dot plots). Probability (*P*) was tested with the parametric unpaired *t*-test (GraphPad Prism 5.04, GraphPad Software, San Diego, CA 92108, USA) compared to PBS vehicle control. Differences were determined as not significant with ns > 0.050 and significant with * *p* < 0.050, ** *p* < 0.010, or *** *p* < 0.001.

**Table 1 cancers-11-00244-t001:** Established spontaneous mouse-derived isograft (sMDI) models, summarizing the characteristics of established sMDIs.

Spontaneous MDI										
**Tumor Name**	**JA-0009**	**JA-0011**	**JA-0013**	**JA-0017**	**JA-0018**	**JA-0023**		**JA-0032**	**JA-0034**	**JA-0021**
						(back)	(kidney)			
**Hematopoietic (y/n)**	no	yes (?)	yes	no	yes	no	no	no	yes	yes
**Histopathological Diagnosis**	low differentiated ADC	HS or HAL	malignant lymphoma	ADC	malignant lymphoma	mammary invasive ADC	clear cell-like ADC of kidney	papillary ADC lung	malignant lymphoma	malignant lymphoma
**Mouse Strain**	DBA/2N	DBA/2N	C57BL/6N albino	CBA/J	CBA/J	C3H/HeJ	C3H/HeJ	BALB/c	C57BL/6N albino	thymus (THY) SCID/bg 1205-16
**Sex**	♀	♀	♀	♀	♀	♀	♀	♂	♂	♀
**Estimated Growth Time (In Days)**	21–23	27–72 problem: liver-/splenomegaly	45	81 to ≥158	60	29–45	60	45–59	45	24
**Included Animals**	growth curve (12) *	frozen/directly (12/10) *** Pool ^§^	frozen (11) **	growth curve (12)	frozen/directly (3/3)	growth curve (12)	frozen/directly (3/4)	growth curve (12)	frozen (12) only gut	frozen/directly (3/6) only SCID/bg
**Take Rate**	92% 11/12	63% 14/22 ^&^	100% 11/11	75% 9/12	100% 6/6	100% 12/12	71% 5/7	66% 8/12	92% 11/12	78% 7/9
***Appearance* (*AP*) Day**	6–7	nd	nd	49–56	nd	13–20	nd	31 - 41	nd	nd
***Real Running Time* (*RRT*) Day** **(% Alive Animals)**	21 (75%) 23 (67%)	nd	nd	81 (58%) 105 (50%)	nd	29 (92%) 45 (42%)	nd	45 (67%) 59 (42%)	nd	nd has to be done in H-2^d^ × H-2^b^ hybrid mouse

Tumors were of non-hematopoietic (no) ADC and hematopoietic (yes) HS/HAL or lymphoma origin from various mouse strains. In the case of tumor JA-0023, two histopathologically different original tumors that were initially back- and kidney-derived were observed. Estimated growth time is the mean (or range) of tumor growth until termination of at least 12 tumor-bearing animals * using frozen tumor pieces (growth curves). In the hematopoietic and the JA-0023 kidney ADC models, ** growth times were calculated from the outgrowth of frozen tumor pieces in various experiments (or if the number of these animals was six or less) of *** frozen or directly re-transplanted tumor pieces. The number of animals is provided in brackets. The take rates reflect the percentage of the animals of respective models developing a tumor within the estimated growth time. In the case of JA-0011, a HS/HAL, a ^§^ pool of primary kidney and gut tumor isolates grown on the left and right flank in the DBA/2 mouse 1307-16 was used. Since several animals did not show local s.c. tumor growth but developed spleno- and/or hepatomegaly causing death, a ^&^ higher number of animals was included to calculate the take rates. In the case of lymphoma JA-0034, suspicious tissues from a different origin grew, but the most stable outgrowth was observed with tissue initially isolated from a gut tumor (used here). *Appearance* is defined as the earliest time point(s) allowing for robust randomization at mean tumor volumes between 40 and 150 mm^3^ in re-transplanted animals. *Real running time* (*RRT*) determines the time difference from implantation to the time point when the remaining animal number reaches ~60% of the starting group size. Therefore, it defines the *treatment time window* (*TTW*), i.e., the maximal time range to treat animals after randomization (*AP*) to the potential study end. This allowed us to calculate the realistic study length as well as necessary group size for statistically reliable analysis for testing the anti-cancer agents. Grey columns are not yet characterized by growth curves (below-6) or the tumor grows only in another mouse strain or because is of other strain origin (1), or differentiates hematopoietic (grey) and non-hematopoietic tumors.

**Table 2 cancers-11-00244-t002:** Mouse strains and sex included in generation of sMDI.

Mouse Strain	CRL Nomenclature	Sex	H-2 Haplotype	Tumor Growth	
				*n*	(*n*)
Black 6	C57BL/6NCrl	♂	H-2^b^	-	(1)
		♀			
Black 6 albino	C57BL/6N-Tyrc-Brd/BrdCrCrl	♂	H-2^b^	1	(1)
		♀		1	(1)
Balb/c	BALB/cAnNCrl	♂	H-2^d^	1	(2)
		♀		-	(1)
CBA/J	CBA/J-Pde6brd1	♀	H-2^k^	2	-
DBA/2	DBA/2NCrl	♀	H-2^d^	2	-
C3H/HeJ	C3H/HeJ-Pde6brd1	♀	H-2^k^	1	(1)
SCID beige	CB17.Cg-PrkdcscidLystbg/Crl	♀	H-2^d^ × H-2^b^	1 *	-

Note: Four animals from each of the specified strains and sexes were included for long term observation. Ten mice were found dead for unknown reasons (*n* = 6) or had to be euthanized for ethical reasons (*n* = 4) other than tumor signs. SCID/bg mice were used as the recipient hosts only to avoid eventual tumor rejection in immunocompetent mice. Tumor growth is reported for *n* = number of animals with finally established sMDI tumors as well as for (*n*) = number of candidates, i.e., potential frozen sMDI samples. * The thymus of one SCID/bg recipient mouse was found to be malignant after re-transplantation. CRL–Charles River Laboratories (Sulzfeld, Germany).

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
