# Peer review of "Mouse-Derived Isograft (MDI) In Vivo Tumor Models I. Spontaneous sMDI Models: Characterization and Cancer Therapeutic Approaches"

_cancers, 2019, doi:10.3390/cancers11020244_

Round 1
Reviewer 1 Report
The authors have designed and developed unique animal models with syngenic mice for various cancers,they have followed the development of cancer and metastsis in these animals for more than a year;histo pathological diagnosis,RNA sequencing and other techniques were employed to verify the soundness of this in vivo system and the authors conclude that these animal model systems could be of value for both diagnostic and therapeutic strategy development for various cancers. The manuscript is very interesting and cold help in developing newer models of human cancer models in mice!however,more studies are needed with more different types of cancers and larger cohort of patient populations for coming to any conclusive support of these strategies to claim it to be more pertinent for cancer therapy than the existing models.
Author Response
sMDI - Author’s Reply to the Review Report - Reviewer1
Dear Madam, dear Sir,
Thank you very much for critical reviewing and your helpful suggestions for revision of our manuscript cancers-409311 “Mouse-Derived Isograft (MDI) in vivo Tumor Models I. Spontaneous sMDI Models: Characterization and Cancer Therapeutic Approaches”.
According to criticism regarding excessive switching between tumor entities in the manuscript by other reviewers, we restructured the presentation of individual tumor entities, and also enhanced cross reference to the presented supplementary data in a more precise manner (see revised manuscript). We also expanded supplementary data files itself with more detailed histopathological analysis of the MDI models.
We hope revised manuscript will possibly improve presentation of the manuscript and find again your acceptance. Thank you for your support.
Kind regards,
Peter
(on behalf of all coauthors)
Reviewer 2 Report
The study carried out in this manuscript is timely and required as better models are needed for pre-clinical studies.
Major Points:
The authors do not refer to the supplementary data in a precise manner which makes it difficult to correlate the data. For example, they just say see supplementary data but do not refer to which one.
While the H&E staining looks convincing immuno-histological analysis is required for each isolated/generated tumor entity to establish their identity. for example CD45, B cell and T cell markers or Diffuse large B cell lymphoma can have germinal center phenotype (CD10+, BCL2+ ), post-germinal center phenotype (CD10−, BCL2− ), Similarly for ADC or mammary invasive ADC.
The authors switch between tumor entities which makes it difficult to follow. I recommend re-writing so that certain sections focus on one of the isografts in each section.
The immunological profiling is carried out predominantly by whole tumor RNA-seq which may not accurately reflect the cell types present. Transformed cells often co-opt immunological markers hence a flow cytometric analysis needs to be shown for all the tumors as done for JA-0009
Enthusiasm for this manuscript is dampened by the lack of orthotopic (at the same site of tumor growth) grafts.
Author Response
sMDI - Author’s Reply to the Review Report - Reviewer2
Dear Madam, dear Sir,
Thank you very much for critical reviewing and your helpful suggestions for revision of our manuscript cancers-409311 “Mouse-Derived Isograft (MDI) in vivo Tumor Models I. Spontaneous sMDI Models: Characterization and Cancer Therapeutic Approaches”.
In response to your review we restructured the presentation of individual tumor entities, and hope it might make it easier to follow. We also enhanced cross reference to the presented supplementary data in a more precise manner (see revised manuscript). We trust this improves readability of the paper.
Your proposal also to include immune-histological analysis especially in case of lymphomas, but also of adenocarcinomas (ADC) has been intensively discussed by the authors, too (and has already been started), but was rejected concerning the present manuscript(s). In our opinion, it would inadequately increase the level of data, since we assume that immune-histological characterization of sMDI as well as cMDI rather represents a separate subject of depiction. It would really enhance the understanding of established MDI models but was not the matter of the manuscript (please see also the statement of our pathologist - below). The main points of our recent presentation are i) generally to describe the idea and implementation of the new MDI tumor model(s) for improved preclinical studies, ii) to show their general properties, and iii) to verify their applicability for testing therapeutic approaches.
Similar arguments apply to your critics regarding immunological profiling carried out predominantly by only whole tumor RNA-seq. It is completely correct that it may not accurately reflect the present cell types and need further analysis to characterize immunological profile. But also here, we present only the general suitability (demonstrated in case of sMDI JA-0009 as well as cMDI JA.2011, and JA-2042) to use RNA-seq as meaningful completion to flow cytometry, and histology.
Your proposal to apply MDI tumor pieces also for orthotopic models sounds highly interesting. We also thought about this point. But there are two main reasons in the moment why we did not include this approach. First, as discussed in the paper, in most cases we do not know if isolated malignant tissue was originated from the place of primary detection (or if it might be of secondary or metastatic origin), i.e. we do not know the tissue of origin. And second (a rather technical problem), an orthotopic application (we are well experienced with the method in standard cell line models) of 1-3 mm3 tumor pieces appears very difficult, at least in some small tissues (e.g. prostate, pancreas, mammary gland etc.), or needs other techniques of tissue attachment (e.g. “by sticking tumor pieces” etc.).
We hope performed revisions as well as our statements will answer your´s expectation. Thank you again for your support.
Kind regards,
Peter
(on behalf of all coauthors)
MDPI CANCER response to reviewers (TL)
Mouse-derived Isografts MDI papers: s-MDI and c-MDI
Dear Reviewers:
The scientific value of integrating immunohistochemical profiling and other more advanced in situ molecular pathology investigations have already been discussed several times amongst the team and we all agree it would have much value to be added later on. However, molecular profiling of the tumor lineage is not the main focus of these reports, although it is obvious these tasks will be undertaken in the future, should these MDI tumors be further utilized.
Diagnosis and phenotypic stability or drift have already been very suitably characterized using standard pathology and routine histochemical HE stains. It is planned to further characterize these lineages not only for infiltrating immune cells using immune cell markers, but also hopefully to further demonstrate tumor cell phenotype stability, should they be further used.
However, at this point of time, as an experience experimental pathologist, I consider that immunophenotyping of tissue sections is not mandatory for presenting this work, and offer limited value for phenotyping, except maybe to further characterize spontaneous lymphoma and carcinogen-induced sarcoma.
We value your feedback.
Best regards,
Thomas Lemarchand, DVM, PhD, dipECVP, DESV-AP.
Reviewer 3 Report
The study is rigorous and useful for investigating spontaneous tumors arisen in aged mice. However its application for drug discovery might be limited. The approach is not innovative but it’s beneficial. However the benefit should be assessed in more details. Tumor characterization and standardization remain a concern. We strongly suggest to the authors to show comparison studies between their models (particularly sMDI JA-0009) and common tumor transplantation models, using syngeneic mice and well established tumor cell lines.
Author Response
sMDI - Author’s Reply to the Review Report - Reviewer3
Dear Madam, dear Sir,
Thank you very much for critical reviewing and your helpful suggestions for revision of our manuscript cancers-409311 “Mouse-Derived Isograft (MDI) in vivo Tumor Models I. Spontaneous sMDI Models: Characterization and Cancer Therapeutic Approaches”.
In response to your review, however, we have to state that the idea of propagating intact tumor pieces in syngeneic mice, rather than using a few established cell lines which are cultured on plastic (or deriving new cell lines) is valuable in terms of having new models with a more originally tumor microenvironment, including immune components, to test immunotherapies and other therapies. In contrast to your view, we and also other reviewers assess the concept not only to be innovative but also potentially both exciting and important. We also think the efficacy data using JA-0009 in this paper (and cMDI JA-2011, JA-2019, JA-2041, and JA-2042 in the accompanying one) clearly show the usability of these models e.g. also for drug discovery or immunotherapeutic approaches.
Your critics regarding tumor characterization and standardization might be result of much switching between tumor entities in the manuscript, also criticized by other reviewers. Thus, we restructured the presentation of individual tumor entities, and also enhanced cross reference to the presented supplementary data in a more precise manner (see revised manuscript), and hope it is now clearer as before. We also expanded supplementary data files with detailed histopathological analysis of the MDI models.
The latter point (unprecise cross reference to supp. data) might also be important regarding your request to compare efficacy data in MDI JA-0009 model with common tumor transplantation models, using syngeneic mice and well established tumor cell lines, since the latter are shown in supplementary data for seven of these models.
We hope revised manuscript will answer also in these points your´s expection for possible improvement of the manuscript. Thank you again for your support.
Kind regards,
Peter
(on behalf of all coauthors)
Reviewer 4 Report
The idea of propagating intact tumor pieces in syngeneic mice, rather than using a few established cell lines which are cultured on plastic (or deriving new cell lines) is valuable in terms of having models with a more normal tumor microenvironment, including immune components, in which to test immunotherapies and other therapies. This group is trying to do this by propagating both spontaneous tumors which arise in older mice (spontaneous mouse-derived isografts= sMDI in paper I). So the concept is potentially both exciting and important. However, sMDI paper #1 has several substantial weaknesses and requires extensive revision before this model can be convincing to others.
Major points:
The figures are out of order. Fig 1b precedes Fig 1a. And in both cases, the micrographs are the size of postage stamps and none of the features detailed in the text can be seen. The photos need to be a lot bigger and the features that are mentioned need to be labeled so the reader can see them. If the conclusion is that the engrafted passages maintain the histology of the original tumor, we need to be convinced of that. And, in particular, the stroma needs to be compared. Further, the pathologist who analyzed these slides should be named in the Methods or elsewhere in the paper.
The descriptions of the origins of the tumors, number of mice, “finalization” of the model etc needs to be clearer. The descriptions between 2.1 establishment and 2.2 Histology/pathology are repetitious, and overall very confusing. Also line 164 says “in models not finished yet” and it’s not quite clear what this means. Why are the authors publishing a paper with models that aren’t finished?
The origin/tumor type of these models is contradictory and apparently not really known, although there are photos of the original tumors in Fig 1. It is stated in line 339 that “we could not yet clearly differentiate the definite tissue origin of most outgrowing MDI tumor models which made it difficult to compare tumor gene expression with its respective, but unfortunately unknown- normal equivalent”. However, later in line 452 in the discussion, they state “Only in the case of sMDIJA-0021 the origin of the tumor remained unclear…”
Comparing 1 MDI model with cell line in terms of immune contexture and response to therapy doesn’t show that this is a better, more useful model. We don’t know if the outcomes are because of just this one particular tumor or because of differences in the model.
The RNAseq section should be removed since there is no useful data there- it is not necessary to show that you can get RNAseq data from these tumor grafts.
Section 2.4 with the flow data has several problems.
What the authors are calling neutrophils are likely gMDSC (CD11b+Ly6g+CD45+). This should be addressed and clarified.
It is important to compare the immune content of the original tumor to several later passages (or just several passages) to show how whether the immune contexture is maintained over multiple passages. Knowing whether this is true would make this a much more characterized model. Also, between tumored mice in the same passage (i.e. see models with divergent growth pattern within a cohort of mice).
c. Results from 4T1 data in which the number of CD8 cells is about 10% seem very high compared to other reports, and the ratio to CD4 cells also seem inconsistent. Can these discrepancies be addressed?
Other points: It is necessary to have extensive editing for proper English usage. Usage of words such as “abortion” when they mean euthanasia and “conspicuous” where “suspicious” might be more appropriate are only two of many examples. There are many other places where sentence construction makes the idea hard to follow, too numerous to list here.
Other corrections needed:
Authors need to carefully proofread the paper: Fig 2a has P 1286-16, whereas text refers to 1280-16.
All abbreviations in figures need to be explained in figure legends.
All instances where sex is mentioned have double symbols= ♀♀ and ♂♂. Please just use 1, double is confusing
In Table 1, please clarify the categories in the 1st column. It is not clear to the reader what “estimated growth time”, “Appearance”, or “Real Running Time” could be. Although these are eventually mentioned in the text, it is well after the mention related to Table 1.
Author Response
sMDI/ cMDI -Author’s Reply to the Review Report - Reviewer4 sMDI and Reviewer2 cMDI
Dear Madam, dear Sir,
Thank you very much for your detailed and critical reviewing as well as your very helpful comments and suggestions for revision of our manuscripts cancers-409311 “Mouse-Derived Isograft (MDI) in vivo Tumor Models I. Spontaneous sMDI Models: Characterization and Cancer Therapeutic Approaches” and cancers-409369 “Mouse-Derived Isograft (MDI) in vivo Tumor Models II. Carcinogen-Induced cMDI Models: Characterization and Cancer Therapeutic Approaches”.
Considering your remarks we tried to remove the substantial weaknesses by extensive revision of our two manuscripts. In the following, I would like to address some of the major topics of your valid and substantial critics. The point to point answers are given in the order concerning manuscript 1#, 2# or both ones.
(manuscript #1)
- wrong arrangement of Figs. 1b and 1a was the result of editorial processing of the manuscript #1. We had overlooked that reference to Fig. 1b preceded reference to Fig. 1a in the manuscript text, and thus the editor sorted the figures in reverse direction. They are now renamed, and in chronological order.
- The descriptions between 2.1 establishment and 2.2 Histology/pathology are repetitious, and overall very confusing: Familiar with the about 3-year long history of establishing the MDI models, our writing and reading of the manuscript(s) was “biased” by the logic of each step gone on this way. To introduce into the strategy and logic of our doing we thought -and I still do so- that a description of MDI establishing along the timeline (like a history) might be helpful to better understand the steps on this way.
Therefore, in chapter 2.1 we provide now a “historical” overview of establishing process while clearly separated chapter 2.2 shows histopathological diagnosis of the various sMDI tumor either of hematopoietic or non-hematopoietic origin.
- Origin of the tumors – tumor type of these models: As accounted now in the text, in most cases, we could not determine the origin of primary tumor/suspicious tissue isolates. This means that we could actually not differentiate if isolates represent the primary tumor or rather secondary metastatic tissues. Tumorous tissue isolated from the lung of JA-0018 CBA/J (lymphoma) or enlarged axillary lymph nodes found in JA-0009 DBA/2N mice (adenocarcinoma) exemplary illustrate the problem. Thus, we would like definitely to point out that in case of sMDI (in cMDI with primary tumors on carcinogen injection sides the situation is another one) the preliminary organ/tissue origin of the tumors only could be verified by histopathological findings.
- sMDI JA-0021: In case JA-0021 the Origin of the tumor also remained unclear for a time. But here, the exclusive growth of the tumor only in SCID/bg mice additionally raised the question in which mouse strain this sMDI originated. The histological finding that primary C3H/HeJ isolated lymph node showed a questionable lymphoma or even non-malignant morphology indicates that malignant tissue might probably originate also from the suspicious, re-
transplanted SCID/bg mouse thymus tissue. This is communicated now more distinct in the text, but exact origin has finally to be clarified.
(manuscript #2)
- regarding “i.p.” route of carcinogen administration in female mice: We started i.p. application only in male animals surprisingly and alarmingly 3/4 animals died within 1-2 days. The fourth one, however, was well and did not show any impairments. Since we really did and still do not have any idea why the animals died, we did not treat the female mice via the i.p. route, i.e. this group was removed from the study.
- I do not understand the difference in tumor growth variability / tumor JA-2019 has an extremely variable growth rate in Fig. 3 - in Fig 5 B shows a very uniform control tumor growth rate: if you compare the data, the mentioned discrepancy between tumor growth variability in figs. 3 and 5 is probably only a perceived one. Whereas in Fig. 3 individual growth curves with different growth length of single mice are shown, Fig. 5 shows the mean curve of the vehicle group with respective SEM values during the treatment period. However, the actual growth heterogeneity in the vehicle group of efficacy study (Fig. 5), is better to realize considering individual tumor volumes at day 21 shown in the right dot plots (ranging from about 150 – 2,800 mm3). A similar picture was seen already on day 15 (not shown) with individual tumor volumes ranging from about 75 – 1,450 mm3.
- How do the authors know that at least some of the effects of anti-checkpoint inhibitors are not due to having randomly included slower growing tumors: We used for randomization 36 animals divided into three groups of 12 animals each, a robust automated random number generation within individual blocks (MS-Excel 2016), which guarantees a more or less, but statistically homogeneous distribution of differently growing tumors in the various groups (Materials and Methods). The necessary group size of 10 – 12 animals was calculated by neutral external statistician based on the growth curves shown in Fig. 3.
- no comparison, or conclusions related to tumor growth rate differences, are given (except for JA-2011 in the Discussion) - What was the point of comparing the cell lines?: At this time point of model characterization broadly varying individual growth time periods were observed in the four cMDI (Tab. 2). In contrast, their real running times (RRT), determining the actual treatment time window (TTW) as well as potential study endpoints are rather similar in untreated mice (Tab. 2, Fig. 5). Since these terms were not defined distinct enough, we changed their presentation and hope they are in an intelligible form now (chapter 2.3).
Since cMDI RRT were also very similar compared to growth curves (RRT) of “other cell line-derived” tumors (Fig. S1– sMD) we looked for common properties of the different models. However, in the seven cell line-based models, neither growth rate (varying from 18 – 25 days), nor hetero- (CT26.wt, Clone M3, or B16.F10) or homogeneity (LL/2, 4T1, RENCA; or MC38-CEA) of growth displayed any relation to the therapeutic outcome with immune checkpoint inhibitors (Fig. S1-sMDI).
Main aspect of comparing the four cMDI (but also sMDI) with cell line-based models, was the idea to compare the new MDI models with as many parameters as possible of the already often used and well characterized cell line models. Thus, we did not consider a relation between “fast and homogeneous growth” and “non-responder” status in cMDI. JA-2011 showed fastest and most homogeneous growth of the four investigated cMDI models. But the resistance of non-responder model JA-2011 to anti-ICPI treatment seems to display another phenomenon.
(manuscripts #1 and #2)
- “All abbreviations in figures need to be explained in figure legends.”: There is an entire abbreviation list at the end of the manuscript, and to my opinion a separate list in each figure would needlessly expand length of the legends. Additionally, each abbreviation in the text is explained with first appearance.
- “established” models / “finalization” / “models not finished yet” / “a paper with models that aren´t finished”: as already mentioned above “writing and reading of the manuscript(s) was “biased” by the logic of each step gone on this way ….”
We now changed and “homogenized” our terminology in the revised manuscripts to exactly describe the status quo of the respective models,
using the following terminology:
term “established” – now defined as: stable outgrowth from frozen tumor pieces, i.e. re-transplantable samples are permanently secured (i.e. not dependent on persistent in vivo amplification).
term “finalized” – now removed: models status quo is described, either i) histopathologically analyzed, ii) with determined growth curves, or iii) further characterized by flow cytometry and/or RNAseq analysis.
term “not (yet) finished” – now removed: these isolates, only primarily re-transplanted and outgrown once or a few times were now only accounted as potential MDI samples in the text or legend of tables.
- the micrographs are the size of postage stamps and none of the features detailed in the text can be seen / and other missing features / pathologist should be named: We summarize the relevant histopathological data as origin of primary isolate, H&E staining of this tissue and of another, s.c. re-transplanted follow-up-sample of the same tumor, together with tumor name, mouse strain, sex, and the tumor diagnoses, in order to submit as much information as possible in one (or two) figures each. I think, in general the figures are very informative, but not unusable to give more detailed information. Since assessment was the sum of several pictures, it could not simply be condensed within one figure by the pathologist. The initials TL of him, Thomas Lemarchand, had been already displayed in “Author Contributions” section.
After intense internal discussion, we decided on the following solution: Small corrections were applied (e.g. removing the double sex symbols) to the figures in the manuscript to still show the general information in one or two figures in the articles. But to allow evaluation in depth of the pathohistological data, we submit largely expanded supplementary data for each
manuscript: high resolution images of different magnification with the relevant features highlighted in the text for the analyzed tumors.
- Section 2.4 with the flow data has several problems.
What the authors are calling neutrophils are likely gMDSC (CD11b+Ly6g+CD45+). This should be addressed and clarified.
Our statement: G-MDSC are characterized as CD11b+, Ly6Clow/neg and Ly6Ghigh phenotype, whereas for neutrophils a rather CD11b+, Ly6G+, Ly6Cint phenotype. Cells gated in our manuscripts as neutrophils, i.e. of CD11b+, Ly6G+, Ly6Cint phenotype, could not definitely be discriminated from G-MDSC of CD11b+, Ly6Clow/neg and Ly6Ghigh phenotype, i.e. both cell types could be contained in the gated population. Our gating was performed by using FMO controls, and we tend from our experience the population rather to be Ly6Cint then Ly6Clow/neg (Figs. 5A-sMDI, and 4A-cMDI), and thus to be neutrophils. However, in general it is very difficult and a matter of debate to discriminate the two populations [1,2]. Also any additional functional testing has not to bring more clarity [1,2]. Thus, we use(d) the term neutrophils to describe this cell population but are aware that it is not the whole truth. Therefore, we added now the information that neutrophils and G-MDSC could be members of the population in the figure legend.
- Results from 4T1 data in which the number of CD8 cells is about 10% seem very high compared to other reports, and the ratio to CD4 cells also seem inconsistent. Can these discrepancies be addressed?
Regarding this point we have to thank you very much for your attentive proof reading, since my colleagues preparing the graph detected a striking CD8+ copy/paste error happened within the graph data processing. We have corrected it in the revised manuscript, so that the number of tumor infiltrating CD8+ goes into normal range, and thus also the CD4/CD8 ratio.
- Your proposal “It is important to compare the immune content of the original tumor to several later passages (or just several passages) to show how whether the immune contexture is maintained over multiple passages. Knowing whether this is true would make this a much more characterized model. Also, between tumored mice in the same passage (i.e. see models with divergent growth pattern within a cohort of mice).”: strikes at the heart, but would -in my opinion- go beyond the scope of these two manuscripts, which are principally describing establishment and general properties of the new MDI models. Thus, I think more detailed analyses of a single MDI, the comparison of various sMDI, cMDI or between them by flow cytometry, RNAseq analyses and e.g. possibly by immuno-histochemistry should be the matter of future investigations.
- The RNAseq section should be removed: We do not agree with you in this point. It is true that not much (useful) data could be presented from the very small window of analyzed expression of various genes in only three gene families by RNAseg. But already the few data demonstrate i) the very different gene expression patterns between two related sarcomas (JA-2011 and JA-2042) but differing in route (s.c. versus i.m.) and carcinogen (MNU versus MCA) used for induction. ii) Data also confirm flow cytometry data regarding various tumor infiltrating
leukocytes by RNAseq, e.g. high CD44 gene expression in M2-macrophage infiltrated JA-0009, or enhanced Cd4 gene expression in JA-2041 with actual enhanced CD4 cell infiltration, which justifies to my opinion the presentation of these results.
Thus, we revised this section to make clear our arguments, and referred more exact to these results (only presented in supplementary data of manuscript #1) also in manuscript #2. We introduced also Figures better summarizing these results. Additionally, we will discuss with the editors to make independently available the supplementary data for both manuscripts each.
- It is necessary to have extensive editing for proper English usage – misspelled - carefully proofread the paper: To improve entire readability we made use of the MDPI “Specialist edit” service. It is clearly better now, but if it was helpful at all – I don´t know – it was hard work again to eliminate some newly introduced mistakes, e.g. as “graft versus the host reaction”.
Thank you again for your work
Kind regards,
Peter
(on behalf of all coauthors)
Round 2
Reviewer 3 Report
The revised manuscript has been significantly improved.
The authors revised their manuscript as recommended.
The quality of the manuscript reached the level to be accepted by Cells.
Author Response
sMDI - Author’s Reply to the Review Report - Reviewer3
Dear Madam, dear Sir,
Excuse the namelessness of my cover note but I did not found any name, although it was ticked off “Open Review ( ) I would not like to sign my review report
(x) I would like to sign my review report”.
First of all, thank you very much for accepting our revised sMDI manuscript 409311.
Additionally, I would like to thank you very much for your detailed and critical reviewing as well as your very helpful comments and suggestions which have significantly improved our revised manuscript.
Kind regards,
Peter
(on behalf of all coauthors)

Reviewer 4 Report
The RNA seq data should be qualified in the Discussion for being a very small sample for analysis.
Author Response
sMDI/ cMDI -Author’s Reply to the Review Report - Reviewer4 sMDI and Reviewer2 cMDI
Dear Madam, dear Sir,
Excuse the namelessness of my cover note but I did not found any name, although it was ticked off “Open Review ( ) I would not like to sign my review report
(x) I would like to sign my review report”.
First of all, thank you very much for accepting revised cMDI manuscript 409369.
Regarding the sMDI manuscript 409311 I/we do well understand your requirement to demonstrate the merely limited significance (i.e. the weakness) of the preliminary RNA-Seq data. But I would very like show also these limited results to include a prospective view on the models into the manuscript. Thus, I have tried now to qualify the data by inserting a short paragraph into the Discussion to be only a very small sample for analysis. Additionally, I added also an “only” in result section 2.5. to emphasize again the preliminarity of these data.
Regarding your suggestion also to enhance “English language and style: (x) Moderate English changes required” I would like to state two points. First, we do not have any native speaker in the group, and we are a small company. Thus, we cannot easily ask native speaking colleagues to help us, as formerly in academic situation. Our customers would probably be confused if asking them. And second, as mentioned already in my last cover letter, we made use of the MDPI “Specialist edit” service. Although, it is (seems to us) clearly better now, your request shows that it was obviously not optimal. This was also our feeling but we could not do it better. Thus, we tried again to enhance language and style in a few points by ourselves, and hope that this might be acceptable.
Finally, I would like to thank you very much again for your detailed and critical reviewing as well as your very helpful comments and suggestions which have significantly improved our revised manuscripts.
Thank you again for your work and support
Kind regards,
Peter
(on behalf of all coauthors)
Peter Jantscheff PhD (group leader)
In Vivo Pharmacology
Proqinase GmbH
Breisacher Str. 117
79106 Freiburg
Germany
peter.jantscheff@t-online.de
Tel: +49-7666-913-0396
